# Exploring the flavor structure of leptons via diffusion models

**Satsuki Nishimura, Hajime Otsuka, and Haruki Uchiyama**

*Department of Physics, Kyushu University, 744 Motooka, Nishi-ku, Fukuoka 819-0395, Japan*
*E-mail:* nishimura.satsuki@phys.kyushu-u.ac.jp,
otsuka.hajime@phys.kyushu-u.ac.jp, uc.haruki496ym@gmail.com

ABSTRACT: We propose a method to explore the flavor structure of leptons using diffusion models, which are known as one of generative artificial intelligence (generative AI). We consider a simple extension of the Standard Model with the type I seesaw mechanism and train a neural network to generate the neutrino mass matrix. By utilizing transfer learning, the diffusion model generates 104 solutions that are consistent with the neutrino mass squared differences and the leptonic mixing angles. The distributions of the CP phases and the sums of neutrino masses, which are not included in the conditional labels but are calculated from the solutions, exhibit non-trivial tendencies. In addition, the effective mass in neutrinoless double beta decay is concentrated near the boundaries of the existing confidence intervals, allowing us to verify the obtained solutions through future experiments. An inverse approach using the diffusion model is expected to facilitate the experimental verification of flavor models from a perspective distinct from conventional analytical methods.

## 1 Introduction

The flavor structure of quarks and leptons is one of the intriguing puzzles in the Standard Model of particle physics. Current experimental and observational data suggest the existence of massive neutrinos and large mixing angles in the lepton sector.

The flavor structure of neutrinos has been addressed in previous works using both top-down and bottom-up approaches. In the top-down approach, we assume specific textures for the Yukawa couplings of charged leptons and neutrinos, which originate from continuous symmetries, such as a $U(1)$ flavor symmetric model using the Froggatt-Nielsen (FN) mechanism [1], non-Abelian discrete symmetries (see for reviews, Refs. [2–8]), modular flavor symmetries [9] (see for reviews, Refs. [10, 11]) and selection rules without group actions [12, 13]. These approaches lead to various textures of neutrino mass matrices at the low-energy scale. Based on the derived textures of Yukawa couplings, one can predict observables such as active neutrino masses, size of CP violation, and lepton flavor violation.

This top-down approach is useful to discuss the connection between low-energy observables and the predictions in flavor symmetric models. On the other hand, in the bottom-up approach as adopted in, for example, Refs. [14–16], they find out a form of neutrino Yukawa couplings that reproduces the experimental data at the low-energy scale. Then, they discuss the lepton flavor violation such as $\text{Br}(\mu \to e\gamma)$ in the framework of supersymmetric models.

The recent developments of machine learning have been utilized in the top-down approach to search for physics beyond the Standard Model, e.g., the flavor structure of quarks and leptons in the $U(1)$ Froggatt-Nielsen model with reinforcement learning [17], the phenomenology of axion model from flavor [18]. In this paper, we adopt a bottom-up approach utilizing generative artificial intelligence (generative AI) to explore the flavor structure of leptons. In particular, we focus on a diffusion model, which is one of the generative models that has been used in high-energy physics, for example, in simulating electron-proton scattering events at the future Electron-Ion Collider [19]. In the context of denoising diffusion probabilistic models (DDPMs) [20], random Gaussian noise is added to a given training data until it approaches pure noise. Subsequently, a neural network is constructed to predict the actual noise that has been added, using these noisy data as input. After the training process, the diffusion model generates new data by denoising through the trained neural network. Furthermore, it is possible to generate desired data from the initial dataset by applying arbitrary conditional labels. This class of diffusion models is referred to as conditional diffusion models.

We employ the DDPM with classifier-free guidance, specifically one of the conditional diffusion models, to investigate the unknown flavor structure of neutrinos within a basis in which the Yukawa matrix of charged leptons and gauge interactions are flavor diagonal. In particular, we study the Standard Model with three right-handed neutrinos where the light neutrino masses are generated through the seesaw mechanism [21–24]. Our training dataset consists of randomly chosen neutrino Yukawa couplings and right-handed neutrino masses in a similar to the analysis of neutrino mass anarchy [25]. The labels are defined as the neutrino mass squared differences and mixing angles in the lepton sector. After learning the diagonalization process of the light neutrino mass matrix, we fix the labels as the observed neutrino mass squared differences and mixing angles in the lepton sector at the $3\sigma$ level. Consequently, we can obtain a sufficient number of neutrino Yukawa couplings and right-handed neutrino masses that reproduce the observed data. These results lead to predictions regarding the effective mass in neutrinoless double beta decay and leptonic CP violation.

This paper is organized as follows. After presenting our setup in Sec. 2.1, the conditional diffusion model is introduced in Sec. 2.2. In this section, we explain the diffusion process to train the neural network in Sec. 2.2.1 and the reverse process to generate new data in Sec. 2.2.2. In addition, we perform transfer learning to improve the quality of theoretical predictions fit to data in Sec. 2.2.3. We show results generated by the diffusion model ion in Sec. 3. Sec. 4 is devoted to the conclusion and future prospects. In Appendix

A, we give the formulation of diffusion models. Furthermore, we explain the details of conditional diffusion models in Appendix B and transfer learning in Appendix C.

## 2 Flavor structure of leptons via the conditional diffusion models

After summarizing the flavor structure of leptons in Sec. 2.1, we implement the neutrino mass matrices to the conditional diffusion models in Sec. 2.2.

### 2.1 Preliminaries

We focus on the Majorana neutrino mass with the seesaw mechanism. The relevant Lagrangian is

$$\mathcal{L} = Y_{i\alpha}^{\nu} \bar{N}_i \ell_\alpha H - \frac{1}{2} M_{ij} \bar{N}_i \bar{N}_j + \text{h.c.}, \tag{2.1}$$

where $\ell_\alpha$, $N_j$, and $H$ respectively denote the left-handed lepton doublet, the Higgs doublet, and the right-handed neutrinos. $Y_{i\alpha}^{\nu}$ and $M_{ij}$ respectively represent Yukawa matrix for neutrinos and Majorana mass matrix for right-handed neutrinos in a basis where the charged lepton Yukawa matrix is diagonal. $\alpha$ denote the lepton flavor indices $\alpha = e, \mu, \tau$. By integrating out heavy $\bar{N}_i$, the mass matrix for active neutrinos is given by

$$(m_\nu)_{\alpha\beta} = \langle H \rangle^2 (Y^\nu)_{\alpha i} (M^{-1})_{ij} (Y^\nu)_{j\beta}, \tag{2.2}$$

with $\langle H \rangle$ being the vacuum expectation value of the Higgs field. A complex-valued symmetric matrix $m_\nu$ can be diagonalized by an unitary matrix $U_{\text{PMNS}}$:

$$U_{\text{PMNS}}^T m_\nu U_{\text{PMNS}} = \text{diag}(m_1, m_2, m_3), \tag{2.3}$$

where $\{m_1, m_2, m_3\}$ are taken as real and positive values. The neutrino mass ordering is given by $m_1 < m_2 < m_3$ for the normal hierarchy and $m_3 < m_1 < m_2$ for the inverted hierarchy.

The mixing matrix $U_{\text{PMNS}}$ can be parameterized by three mixing angles $\theta_{ij}$, a CP-violating Dirac phase $\delta_{\text{CP}}$ and two Majorana phases $\alpha_{21}, \alpha_{31}$:

$$U_{\text{PMNS}} = \begin{pmatrix} c_{12}c_{13} & s_{12}c_{13} & s_{13}e^{-i\delta_{\text{CP}}} \\ -s_{12}c_{23} - c_{12}s_{23}s_{13}e^{i\delta_{\text{CP}}} & c_{12}c_{23} - s_{12}s_{23}s_{13}e^{i\delta_{\text{CP}}} & s_{23}c_{13} \\ s_{12}s_{23} - c_{12}c_{23}s_{13}e^{i\delta_{\text{CP}}} & -c_{12}s_{23} - s_{12}c_{23}s_{13}e^{i\delta_{\text{CP}}} & c_{23}c_{13} \end{pmatrix} \begin{pmatrix} 1 & 0 & 0 \\ 0 & e^{i\frac{\alpha_{21}}{2}} & 0 \\ 0 & 0 & e^{i\frac{\alpha_{31}}{2}} \end{pmatrix}. \tag{2.4}$$

Here, the mixings $c_{ij} = \cos\theta_{ij}$ and $s_{ij} = \sin\theta_{ij}$ with $i, j = 1, 2, 3$ and $i < j$ are rewritten as

$$\sin^2\theta_{13} = |(U_{\text{PMNS}})_{13}|^2, \quad \sin^2\theta_{23} = \frac{|(U_{\text{PMNS}})_{23}|^2}{1 - |(U_{\text{PMNS}})_{13}|^2}, \quad \sin^2\theta_{12} = \frac{|(U_{\text{PMNS}})_{12}|^2}{1 - |(U_{\text{PMNS}})_{13}|^2}. \tag{2.5}$$

A rephasing-invariant measure of CP violation is given by the Jarlskog invariant, $J_{\text{CP}}$ from PMNS matrix elements:

$$J_{\text{CP}} = \text{Im}[U_{e1} U_{\mu2} U_{e2}^* U_{\mu1}^*] = s_{23} c_{23} s_{12} c_{12} s_{13} c_{13}^2 \sin\delta_{\text{CP}}, \tag{2.6}$$

with $U_{\alpha i} := (U_{\mathrm{PMNS}})_{\alpha i}$, and the Majorana phases are also estimated by other invariants $I_1$ and $I_2$:

$$I_1 = \mathrm{Im}[U_{e1}^* U_{e2}] = c_{12} s_{12} c_{13}^2 \sin\left(\frac{\alpha_{21}}{2}\right), \ \ I_2 = \mathrm{Im}[U_{e1}^* U_{e3}] = c_{12} s_{13} c_{13} \sin\left(\frac{\alpha_{31}}{2} - \delta_{\mathrm{CP}}\right). \tag{2.7}$$

Experimental values for the neutrino mass squared differences, mixing angles, and Dirac CP violation are summarized in Table 1 and the PMNS matrix at the $3\sigma$ CL range is of the form:

$$|U_{\mathrm{PMNS,\,exp}}|_{3\sigma} = \begin{pmatrix} 0.801 \to 0.842 & 0.519 \to 0.580 & 0.142 \to 0.155 \\ 0.252 \to 0.501 & 0.496 \to 0.680 & 0.652 \to 0.756 \\ 0.276 \to 0.518 & 0.485 \to 0.673 & 0.637 \to 0.743 \end{pmatrix}. \tag{2.8}$$

The sum of neutrino masses is constrained as $\sum_i m_i < 0.12$ eV (95%) from the data of Cosmic Microwave Background (CMB) [26] which will be further constrained as $0.059\,\mathrm{eV} < \sum_i m_i < 0.113$ eV at 95% confidence [27] by combining the data from the Dark Energy Spectroscopic Instrument with CMB data.

| Observables | Normal Ordering (NO) | | Inverted Ordering (IO) | |
|---|---|---|---|---|
| | $1\sigma$ range | $3\sigma$ range | $1\sigma$ range | $3\sigma$ range |
| $\sin^2\theta_{12}$ | $0.308^{+0.012}_{-0.011}$ | $0.275 \to 0.345$ | $0.308^{+0.012}_{-0.011}$ | $0.275 \to 0.345$ |
| $\sin^2\theta_{13}$ | $0.02215^{+0.00056}_{-0.00058}$ | $0.02030 \to 0.02388$ | $0.02231^{+0.00056}_{-0.00056}$ | $0.02060 \to 0.02409$ |
| $\sin^2\theta_{23}$ | $0.470^{+0.017}_{-0.013}$ | $0.435 \to 0.585$ | $0.550^{+0.012}_{-0.015}$ | $0.440 \to 0.584$ |
| $\delta_{\mathrm{CP}}/\pi$ | $1.18^{+0.14}_{-0.23}$ | $0.69 \to 2.02$ | $1.52^{+0.12}_{-0.14}$ | $1.12 \to 1.86$ |
| $\dfrac{\Delta m_{21}^2}{10^{-5}\,\mathrm{eV}^2}$ | $7.41^{+0.21}_{-0.20}$ | $6.82 \to 8.03$ | $7.42^{+0.21}_{-0.20}$ | $6.82 \to 8.04$ |
| $\dfrac{\Delta m_{3l}^2}{10^{-3}\,\mathrm{eV}^2}$ | $2.507^{+0.026}_{-0.027}$ | $2.427 \to 2.590$ | $-2.486^{+0.025}_{-0.028}$ | $-2.570 \to -2.406$ |

**Table 1:** Experimental values for the neutrino mass differences, mixing angles and CP phase obtained from global analysis of NuFIT 6.0 with Super-Kamiokande atmospheric data [28], where $\Delta m_{3l}^2 \equiv \Delta m_{31}^2 = m_3^2 - m_1^2 > 0$ for NO and $\Delta m_{3l}^2 \equiv \Delta m_{32}^2 = m_3^2 - m_2^2 < 0$ for IO.

The effective Majorana neutrino mass $m_{\beta\beta}$ for the neutrinoless double beta decay is given by

$$\langle m_{\beta\beta} \rangle = |m_1 c_{12}^2 c_{13}^2 + m_2 s_{12}^2 c_{13}^2 e^{i\alpha_{21}} + m_3 s_{13}^2 e^{i(\alpha_{31} - 2\delta_{\mathrm{CP}})}|, \tag{2.9}$$

which is upper bounded by $0.036$ eV (90% CL) [29]. Moreover, the effective mass of electron neutrino probed by tritium beta decay is defined as follows:

$$m_{\nu_e} = \left(m_1^2 c_{12}^2 c_{13}^2 + m_2^2 s_{12}^2 c_{13}^2 + m_3^2 s_{13}^2\right)^{1/2}, \tag{2.10}$$

which is upper bounded by $0.45$ eV (90% CL) [30].

## 2.2 Conditional diffusion models

In this section, we introduce the conditional diffusion model used in our analysis. That is constructed of a basic diffusion model and classifier-free guidance (CFG). We refer the reader to Appendix A for a formulation of the diffusion models based on Denoising Diffusion Probabilistic Models (DDPMs) and to Appendix B for the detail of the CFG.

The diffusion model consists of two stages, referred to as the diffusion process and the reverse process. In the diffusion process, random noise $\epsilon$ is added to input data $G$, and a machine learns to predict the added noise. In the reverse process, noise $\epsilon_\theta$ output by the machine is gradually removed from the pure Gaussian noise to produce meaningful data. To predict the noise, we utilize neural network models. In a neural network, a $n$-th layer with $N_{n-1}$ dimensional vector $\vec{X}_{n-1} = (X_{n-1}^1, X_{n-1}^2, \ldots, X_{n-1}^{N_{n-1}})$ transforms into a $N_n$ dimensional vector $\vec{X}_n = (X_n^1, X_n^2, \ldots, X_n^{N_n})$ through following way:

$$X_n^i = h_n(w_n^{ij} X_{n-1}^j + b_n^i). \tag{2.11}$$

Here, $h$ is the activation function, $w$ is the weight, and $b$ is the bias respectively. Since the activation function is generally a nonlinear function, the neural network acts on multiple nonlinear transformations. In our analysis, fully-connected layers are considered. In the following, we implement the Standard Model with three right-handed neutrinos to the diffusion model, where the diffusion and reverse processes are respectively described in Sec. 2.2.1 and Sec. 2.2.2.

### 2.2.1 Diffusion process

In the diffusion process, let us define the data sequence $\{x_1, x_2, \ldots, x_T\}$ associated with an initial data $x_0$:

$$x_t = \sqrt{\bar{\alpha}_t} x_0 + \sqrt{\bar{\beta}_t} \epsilon, \tag{2.12}$$

with $t = 1, \ldots, T$. Here, $\epsilon$ obeying a standard normal distribution $\mathcal{N}(0, 1)$ is called a noise, and $\bar{\alpha}_t, \bar{\beta}_t$ are determined as follows:

$$\bar{\alpha}_t = \prod_{s=1}^{t} \alpha_s, \qquad \bar{\beta}_t = 1 - \bar{\alpha}_t, \tag{2.13}$$

with

$$\alpha_t = 1 - \beta_t, \qquad \beta_t = \left(1 - \frac{t}{T}\right)\beta_{\min} + \frac{t}{T}\beta_{\max}, \tag{2.14}$$

where $\alpha_t, \beta_t$ are called noise schedules *. Throughout our analysis, we adopt $\beta_{\min} = 10^{-4}$, $\beta_{\max} = 0.02$, and $T = 1000$. Note that although we used the linearly increasing $\beta_t$ in our

---

*Among the various types of generative AI, the diffusion models with CFG are superior in that it can easily and accurately generate data based on conditional labels $L$. In the context of image generation, various paintings are collected as input data $G$ and associated information as $L$ is also prepared. The inputs $G$ correspond to added noises according to the noise schedules. Based on these data, a machine that has learned to predict noise will be able to output motifs and painting styles specific to an artist. As a result, for example, given a command "make an image of 'The Persistence of Memory' by Salvador Dalí with a touch of Van Gogh", the title of art or the names of artists function as conditional labels $L$, and an image that combines the motif "melting clock" and Van Gogh's "The starry night" is output.

analysis, we could consider a noise schedule that varies according to cosine [31].

We consider the diagonal basis of Majorana mass matrix such as $M = \text{diag}\,(M_1, M_2, M_3)$, where $M_1, M_2$ are complex numbers and $M_3$ is a real number. Then, in preparing the initial data, we deal with the following values:

$$G = \left\{ \text{Re}\, Y_{i\alpha}^{\nu},\ \text{Im}\, Y_{i\alpha}^{\nu},\ \frac{|M_1|}{M_3},\ \frac{|M_2|}{M_3},\ \log_{10}\left( \frac{\langle H \rangle^2}{M_3}\,[\text{eV}] \right),\ \arg M_1,\ \arg M_2 \right\}, \tag{2.15}$$

$$L = \left\{ \log_{10}\left( \Delta m_{21}^2\,[\text{eV}] \right),\ \log_{10}\left( \Delta m_{31}^2\,[\text{eV}] \right),\ |(U_{\text{PMNS}})_{ij}| \right\}, \tag{2.16}$$

with $i, j = 1, 2, 3$. $G$ is a main input in the diffusion model and indicates the set of variables that are subject to data generation. On the other hand, $L$ is an auxiliary label for conditional generation and is passed to the neural network in the appropriate way described later. Now, $G$ has 23 components and $L$ has 11 components in terms of real numbers. Each element of $G$ was generated by uniform random numbers within the following ranges:

$$-1 \leq \{\text{Re}\, Y_{i\alpha}^{\nu},\ \text{Im}\, Y_{i\alpha}^{\nu}\} \leq 1, \quad 10^{-5} \leq \left\{ \frac{|M_1|}{M_3},\ \frac{|M_2|}{M_3} \right\} \leq 1, \tag{2.17}$$

$$-\pi \leq \{\arg M_1,\ \arg M_2\} \leq \pi, \quad -1.22 \leq \log_{10}(\langle H \rangle^2 / M_3\,[\text{eV}]) \leq 2.78. \tag{2.18}$$

In particular, Eq. (2.18) means $6.05 \times 10^{-2}\,\text{eV} \leq \langle H \rangle^2 / M_3 \leq 6.05 \times 10^2\,\text{eV}$, i.e., $10^{14}\,\text{GeV} \leq M_3 \leq 10^{18}\,\text{GeV}$. Moreover, we imposed $|M_1|/M_3 \leq |M_2|/M_3$. Then, $L$ is calculated based on Eq. (2.3) from a random generated $G$. In actual learning, we prepared 100,000 pairs of $(G, L)$. Of these data, 90% are used as training data and 10% as validation data.

The training flow of the neural network in our diffusion model is shown in Fig. 1. When the integer $t$ is randomly selected from $[1, T]$, the noisy data $x_t$ is determined according to Eq. (2.12). When an input layer of the neural network receives noisy pare $(x_t, L)$ as input $X_0$ in Eq. (2.11), it performs nonlinear transformations according to that equation. An output of the network $\epsilon_\theta$ is calculated as the predicted noise, and an error with the added noise $\epsilon$ is evaluated by a loss function. The parameters $\theta = w, b$ of the network are updated by back-propagation based on the value of the loss function, and a well-trained network can accurately estimate the noise component in $x_t$.

To make a conditional diffusion model, we give hidden layers the labels $L$ and the time $t$ in probability of 90%. On the other hand, in 10%, we dropped out the labels by setting $L = \varnothing$. In our program, 0 is used as $\varnothing$.

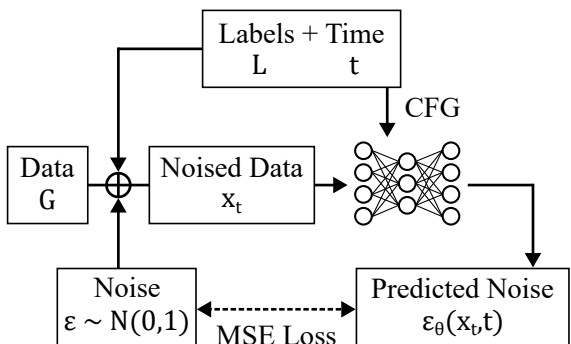

**Figure 1:** The summary of input/output of a neural network in the diffusion process. Based on the noise schedule, noise $\epsilon$ sampled from the standard normal distribution $\mathcal{N}(0,1)$ is added to the data $G$. The inputs of the network are the noisy data $x_t$ and the label information $\{L,t\}$, while the output is the predicted noise $\epsilon_\theta$. In particular, $\{L,t\}$ is also input to the intermediate layers for conditional learning. The network is updated to minimize a mean squared error (MSE) defined from the actual added noise $\epsilon$ and predictive noise $\epsilon_\theta$.

The detailed architecture of our neural network is shown in Table 2. The activation function $h$ in Eq. (2.11) is chosen as a SELU function for hidden layers H1 to H6, a tanh function for a hidden layer H7, and an identity function for an output layer. Since the diffusion model is used to predict noise sampled from a normal distribution, H7 contributes to reduce the divergence by limiting the range of output values with tanh. In addition, a loss function is chosen as the MSE, and we use the ADAM optimizer in PyTorch to update the weights $w$ and biases $b$ in Eq. (2.11). A learning rate is automatically adjusted through the scheduler "OneCycleLR" provided by PyTorch, and we adopt $r_{\max} = 0.001$ as a maximum rate. Then, a batch size is 64 and an updating of the parameter is repeated 100,000 times.

| Layer | Input | H1 | H2 | H3 | deepest H4 |
|---|---|---|---|---|---|
| Dimension | $\mathbb{R}^{21}_x + \mathbb{R}^{12}_{L,t}$ | $\mathbb{R}^{64} + \mathbb{R}^{12}_{L,t}$ | $\mathbb{R}^{128} + \mathbb{R}^{12}_{L,t}$ | $\mathbb{R}^{256} + \mathbb{R}^{12}_{L,t}$ | $\mathbb{R}^{512} + \mathbb{R}^{12}_{L,t}$ |

| Layer | deepest H4 | H5 | H6 | H7 | Output |
|---|---|---|---|---|---|
| Dimension | $\mathbb{R}^{512} + \mathbb{R}^{12}_{L,t}$ | $\mathbb{R}^{256} + \mathbb{R}^{12}_{L,t}$ | $\mathbb{R}^{128} + \mathbb{R}^{12}_{L,t}$ | $\mathbb{R}^{64} + \mathbb{R}^{12}_{L,t}$ | $\mathbb{R}^{21}$ |

**Table 2:** In the neural network, the inputs are the noised data $x_t$ (21 dimensions), the label $L$ (11 dimensions), and the time $t$. The activation functions are chosen as the SELU function for hidden layers H1 to H6. On the other hand, H7 has the tanh function and the output layer has the identity function. Note that the label $L$ is treated as $\varnothing$ with 10% probability.

### 2.2.2 Reverse process

In the reverse process, the generation of the data set is performed with labels that accurately reflect reality. Specifically, $L$ is designated based on the central values in Eq. (2.8) and

Table 1:

$$\log_{10}\left(\Delta m_{21}^2\,[\text{eV}]\right) = \log_{10}\left(7.49 \times 10^{-5}\right) = -4.13, \tag{2.19}$$

$$\log_{10}\left(\Delta m_{31}^2\,[\text{eV}]\right) = \log_{10}\left(2.51 \times 10^{-3}\right) = -2.60, \tag{2.20}$$

$$|U_{\text{PMNS}}| = \begin{pmatrix} 0.822 & 0.550 & 0.149 \\ 0.377 & 0.588 & 0.704 \\ 0.397 & 0.579 & 0.690 \end{pmatrix}. \tag{2.21}$$

Here, we consider the structure of neutrino masses as the normal ordering. The diffusion model can generate data based on the inverted ordering by adjusting $L$ to the experimental values for that scenario, but we only focus on the normal ordering in this paper for simplicity. We will hope to report on this analysis for future work.

From the generated $G$, the label $L$ is recalculated, and the results can be compared to the experimental values. If the obtained result is within a certain error range as described later, the data $G$ reproducing the observables has been generated solely from the experimental results.

### 2.2.3 Transfer learning

When the diffusion process has been completed once using entirely random training data, various new data $G$ can be generated through the reverse process. However, since the original random data are inadequate for accurately reproducing the experimental observables, the current network cannot achieve sufficient accuracy. To improve the accuracy of the reproduction of experimental values by the diffusion model, we perform the transfer learning. A detailed explanation of the transfer learning is provided in Appendix C.

After the data $G$ is generated by the diffusion model, the physical values $P_\ell$ calculated by $G$ can be obtained. The accuracy of $P_\ell$, in comparison to the target label $L$ constructed by experimental values, is quantitatively evaluated by the $\chi^2$ function defined as follows:

$$\chi^2 = \sum_{\ell=1}^{n} \left(\frac{P_\ell - \mu_\ell}{\sigma_\ell}\right)^2, \tag{2.22}$$

where $P_\ell$ is prediction for physical observables, $\mu_\ell$ is a central value and $\sigma_\ell$ is a $1\sigma$ deviation.

Now, we refer to the first neural network which has been trained once as a *pre-network*. For the transfer learning, we collect the data that satisfy $\chi^2 < 5.5 \times 10^3$ using the pre-network. Note that this condition alone would include a lot of data in which both the mass squared differences of neutrinos and mixing angles have low accuracy. In order to improve the efficiency of learning, we collect data in which at least one of the two has high accuracy. Therefore, the following two conditions are taken into account:

1. Mass condition : $\chi^2 < 10^3$ is satisfied only for $P_\ell = \{\Delta m_{21}^2, \Delta m_{31}^2\}$.

2. Mixing condition : $\chi^2 < 4.5 \times 10^3$ is satisfied only for $P_\ell = \{|(U_{\text{PMNS}})_{ij}|\}$.

We respectively prepare 7,289 models and 36,529 models that satisfy the mass condition and the mixing condition. The number of models satisfying both conditions is counted as 78,488. Thus, a new training data is constructed from the 116,306 models. Based on this new data, the pre-network was trained once again. Here, hyper-parameters are the same as those of the first learning. Additionally, all parameters of the network are updated, so this second training phase is referred to as fine-tuning. In this work, the second network constructed from the pre-network is referred to as a *tuned-network*.

## 3   Results

Let us use the tuned-network to generate $3 \times 10^7$ data $G$. Then, we compute the corresponding label $L$ for each of these generated results and compare it to the experimental values in Eq. (2.8) and Table 1. It turns out that there exist 104 solutions such that all physical quantities in $L$ are within the $3\sigma$ confidence interval. Note that the solutions obtained here are combinations of numbers that were not included in the training data and that the diffusion model does not select plausible candidates from the random data prepared in advance. In other words, the diffusion model is autonomously generating a new $G$ that reproduces the real observables given as the label $L$.

We check how much hierarchy occurred in $\{\mathrm{Re}\, Y_{i\alpha}^\nu, \mathrm{Im}\, Y_{i\alpha}^\nu\}$ in the generated data $G$. There are 49, 50, 3, and 2 Yukawa matrices with a total of 18 values having 1, 2, 3, and 4 digits of hierarchy, respectively. This means that 95.2% of the data are within a 2-digit hierarchy. Indeed, in the initial training data in which uniform random numbers were generated in the range of $[-1, 1]$, the percentage of data that fell within a hierarchy of 2 digits was about 94.8%. Therefore, it can be considered that even in the tuned-network that have undergone transfer learning, the uniformity of the initial data is reflected in the generating process. Of course, it is difficult to reproduce experimental values for the mass squared differences of neutrinos and mixing angles within $3\sigma$ simply by generating random numbers. By utilizing the diffusion model, Yukawa matrices can be generated that selectively reproduce the desired physical observables.

Fig. 2 shows the distribution of right-handed neutrino masses $M_1, M_2, M_3$. Since we impose the relation $M_1 \leq M_2 \leq M_3$, the region which do not satisfy this relation is drawn as the gray background in the figure. Although the initial training data is prepared from a wide range of masses, such as Eqs. (2.17) and (2.18), the masses generated by the tuned-network ranged from $10^{15}\,\mathrm{GeV}$ to $10^{16}\,\mathrm{GeV}$. Given that the Yukawa couplings obtained by the tuned-network follow a distribution similar to that of the initial data used in the pre-network, it seems natural that the right-handed neutrino masses should also be distributed over a wide range. However, in the diffusion model constructed in this study, it turns out that the machine autonomously generates the mass scale around $10^{16}\,\mathrm{GeV}$ as a favorable scale to satisfy the existing $3\sigma$ constraints. Furthermore, a strong linear correlation appears in the scatter plots of $M_1$ and $M_3$. Hence, the generated values are not simply chosen as random numbers but must be output based on some relations learned by the neural network.

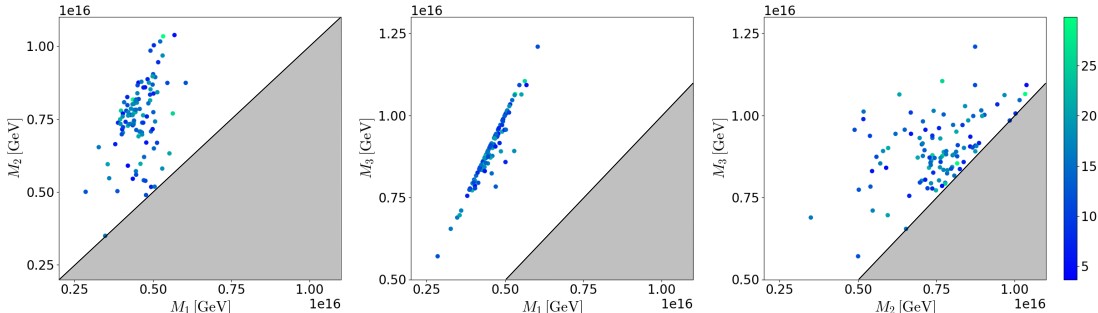

**Figure 2:** The distribution of absolute values of right-handed neutrino masses. The color bar shows $\chi^2$ values with $P_l = \{\Delta m_{21}^2, \Delta m_{31}^2, s_{12}^2, s_{23}^2, s_{13}^2\}$ in Eq. (2.22). In addition, the white region is allowed by the relation $M_1 \leq M_2 \leq M_3$, but the gray region does not satisfy it.

We will now consider what implications these solutions provide for other physical values. Fig. 3 shows the correlation between the Dirac CP phase, the mixing angle $\theta_{23}$ and the sum of neutrino masses. The median value of the sum is $60.3\,\text{meV}$, and the solutions found by the diffusion model are clustered around this median value. Moreover, the Dirac CP phase tends to appear around 106 deg and 228 deg as first and third quartiles of the 104 data respectively, while there are few solutions around 0 deg or 180 deg. This indicates that the lepton sectors exhibit a significant CP violation in order to explain the existing experimental results.

We also check the results for the Majorana phases $\alpha_{21}, \alpha_{31}$. Although $\alpha_{21} = 0\,[\text{deg}]$ is calculated for all of the solutions generated by the diffusion model, $\alpha_{31}$ is found to be distributed over a wide angle. Its distribution is shown in Fig. 4. In particular, no candidates are found near $\alpha_{31} = 0\,[\text{deg}]$. This suggests that CP violation is easily realized in terms of $\alpha_{31}$. We also examine correlations with $\theta_{13}, \theta_{23}$, and the sum of neutrino masses for $\alpha_{31}$, but no significant correlations are found.

Fig. 5 shows the distribution of the effective Majorana neutrino mass $m_{\beta\beta}$ as functions of the sum of neutrino masses and the electron neutrino mass. The region of the 95% CL with an assumption of the normal ordering is shown with a white background based on NuFIT 6.0 (Ref. [28]), and all the solutions generated by the diffusion model fall within this range. In particular, the red lines indicate the boundaries of the 95% CL, and the data are plotted along that outline. Hence, these solutions could be verified through future experiments. In addition, the solutions with small $\chi^2$ values tend to be concentrated in the region of $2\,[\text{meV}] \leq m_{\beta\beta} \leq 4\,[\text{meV}]$. These tendencies seem to emerge as the diffusion model learns some features to reproduce the experimental results. Since it is expected that extracting the features from the neural network will deepen our understanding of the lepton sectors, one of our future tasks is to analyze the data with enhanced interpretability referring to explainable AI.

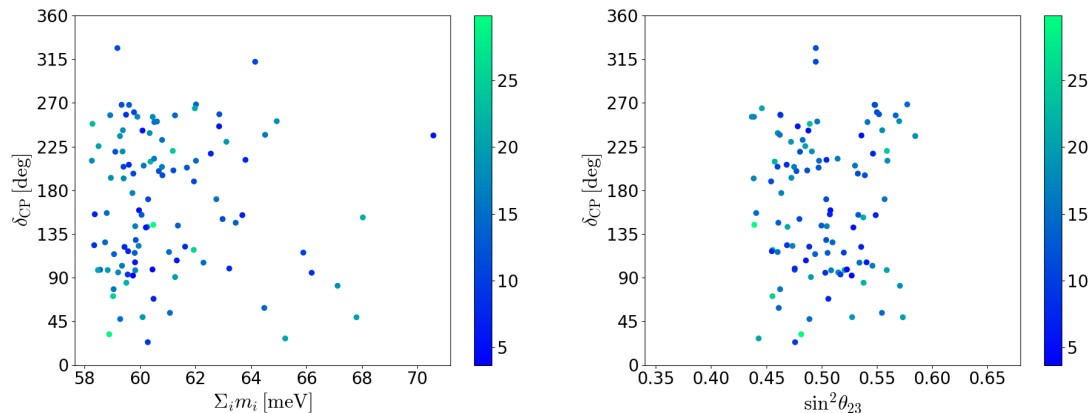

**Figure 3:** The distribution of the Dirac CP phase $\delta_{\rm CP}$ with respect to sum of neutrino masses in the left panel and the mixing angle $\theta_{23}$ in the right panel. The color bars show $\chi^2$ values with $P_l = \{\Delta m^2_{21}, \Delta m^2_{31}, s^2_{12}, s^2_{23}, s^2_{13}\}$ in Eq. (2.22). The solutions are concentrated around $\delta_{\rm CP} = 106, 228$ [deg] and $\Sigma_i m_i = 60.3$ [meV]. Note that for the mixing angle in the right panel, all of the points are located within $3\sigma$ range because we extract the cases that satisfy the experimental constraints from the data generated by the diffusion model.

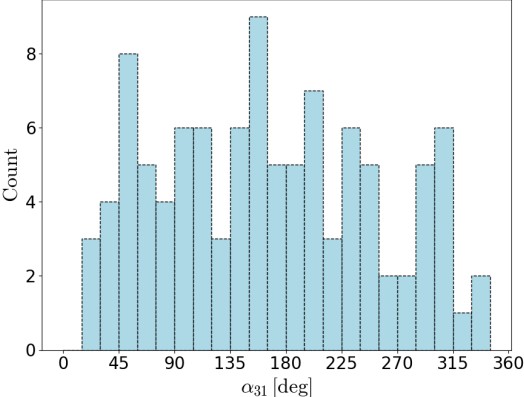

**Figure 4:** The histogram shows the distribution of the Majorana phase $\alpha_{31}$. It turns out that none of the solutions are found near $\alpha_{31} = 0$ [deg].

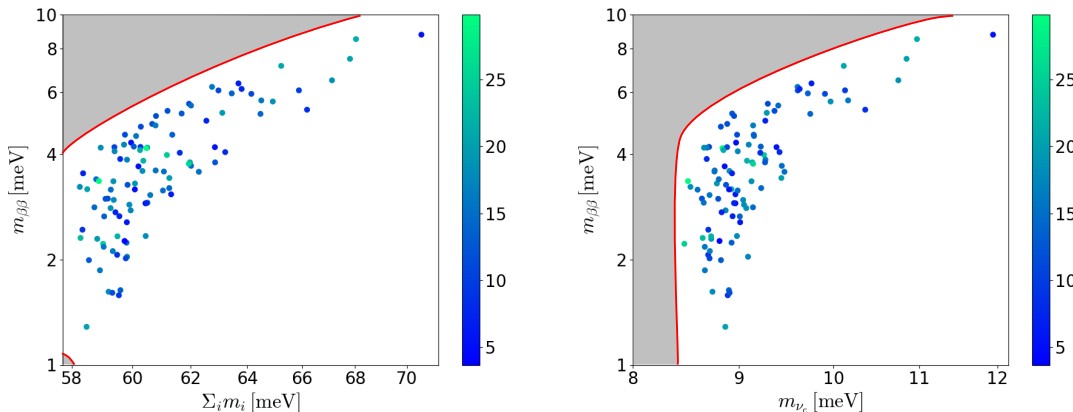

**Figure 5:** The distribution of the effective Majorana neutrino mass $m_{\beta\beta}$ with respect to the sum of neutrino masses in the left panel and the mass of electron neutrino in the right panel. The color bars show $\chi^2$ values with $P_l = \{\Delta m_{21}^2, \Delta m_{31}^2, s_{12}^2, s_{23}^2, s_{13}^2\}$ in Eq. (2.22). The white area is the 95% CL allowed regions for the normal ordering, and the gray area separated by the red boundary means outside of those regions based on NuFIT 6.0 [28]. All the solutions generated by the diffusion model satisfy the restrictions from experiments along the boundaries.

## 4 Conclusions

From various perspectives, numerous efforts have been made to comprehend the flavor structure of the quarks and the leptons in a unified framework. In particular, the nature of neutrinos continues to be enigmatic in many respects. We have considered a simple model with the type I seesaw mechanism and attempted to generate mass matrices that reproduce the experimental results through the diffusion model, known as a type of generative AI. The diffusion model using real experimental values as conditional labels provides a new approach to investigating the distribution and correlation of physical quantities that are expected to be confirmed through future experiments.

In this paper, we have constructed the diffusion model with CFG to explore the flavor structure of leptons. We assumed the type I seesaw mechanism and prepared various mass matrices for the lepton sector based on uniformly distributed random numbers, as detailed in Sec. 2.1. During the diffusion process described in Sec. 2.2.1, the neural network was trained to predict noise, which was added to the initial data, using actual noisy data and conditional labels. Then, we generated new mass matrices of neutrinos utilizing real observables as conditional labels in Sec. 2.2.2, and performed the transfer learning to improve the quality of theoretical predictions fit to data in Sec. 2.2.3. The tuned-network generated the 104 solutions that satisfy the $3\sigma$ constraints of neutrino masses and mixing angles. In Sec. 3, we analyzed the characteristics of the generated results and discovered that the CP phases and the sums of neutrino masses, which were not included in the conditional labels, exhibited non-trivial tendencies. We also showed that the effective Majorana neutrino masses calculated from the generated data are concentrated near the

boundaries of the existing confidence intervals, allowing us to compare the validity of the obtained solutions through future experiments.

Before closing our paper, we will mention possible future works:

- In this study, a straightforward model with only the type I seesaw mechanism was employed to study the CP violation, the effective Majorana neutrino mass, etc. For future analysis, as an example, we can consider lepton flavor violation in theories beyond the Standard Model and try to predict the decay branching ratio based on existing experimental results. Since the analysis with the diffusion models is applicable regardless of the details of models, it is useful for obtaining various predictions from existing observables.

- Within the scope of this study, it has not been examined whether there are correlations between the values of the Yukawa matrices generated by the diffusion model. In particular, if there is any symmetry in the Yukawa matrix, it is derived as a favorable structure to reproduce existing experimental results. In conventional perspective, the analysis of flavor models often begins with the assumption of some symmetry. Conversely, an inverse approach utilizing the diffusion model or other generative AI techniques is expected to facilitate the verification of the flavor models from a different point of view.

- When we generate data using DDPM with the neural network trained on $T = 1000$, it is necessary to perform 1,000 steps of denoising for each data $G$. For large-scale generation, reducing the number of steps is desirable to improve calculational speed. While DDPM is based on stochastic denoising, Ref. [32] proposes Denoising Diffusion Implicit Models (DDIMs), which employ a deterministic generation process. It has been reported that DDIM can achieve accurate image generation while decreasing the number of inference steps. In the previous work, both DDPM and DDIM have been implemented in a unified framework with a single parameter. Additionally, Ref. [33] describes an approach to improve the calculational speed by integrating mechanisms from Generative Adversarial Networks (GANs), another type of generative AI, into the diffusion models. Furthermore, Ref. [34] proposes a method called model-guidance, which has demonstrated accuracy surpassing that of the CFG. Although the diffusion models have achieved remarkable results in image generation, it remains uncertain whether enhanced methods like DDIM will be effective when the neural networks learn eigenvalue decomposition to obtain conditional labels with the hierarchical structure. Comparative studies are needed to confirm whether the latest developments in diffusion models are also valid for the analysis of the flavor models.

## Acknowledgments

This work was supported in part by Kyushu University's Innovator Fellowship Program (S. N.), JSPS KAKENHI Grant Numbers JP23H04512 (H.O).

# A  Formulation of diffusion models

In this section, we introduce the denoising diffusion probabilistic model (DDPM), which is one of the formulations of the diffusion model by Ref. [20]. The DDPM consists of a diffusion process that gradually adds noise to input data and a reverse process that conversely removes the noise. This framework provides an intuitive definition of the diffusion models. When a variable is explicitly specified in our notation, $\mathcal{N}(x; \mu, \sigma)$ is a normal distribution with a random variable $x$, an average $\mu$, and a variance $\sigma$. Furthermore, a conditional probability $P(A|B)$ denotes the probability of $A$ given the condition $B$.

## A.1  Diffusion process and reverse process

Consider the following Markov process that adds noise to an original input data $x_0$ to obtain a series of new data $x_1, x_2, \ldots, x_T$:

$$q(x_{1:T}|x_0) = \prod_{t=1}^{T} q(x_t|x_{t-1}), \tag{A.1}$$

$$q(x_t|x_{t-1}) = \mathcal{N}(x_t; \sqrt{\alpha_t}x_{t-1}, \beta_t), \tag{A.2}$$

with $x_{i:j} = x_i, x_{i+1}, \ldots, x_j$. Here, the parameters $0 < \beta_1 < \beta_2 < \cdots < \beta_T < 1$ decide the variance of $\mathcal{N}$, and $\alpha_t$ is defined as $\alpha_t = 1 - \beta_t$. Thus, $\{\alpha_1, \ldots, \alpha_T, \beta_1, \ldots, \beta_T\}$ are called noise schedules. By the definition of $\alpha$ and the magnitude of $\beta$, the amount of original input data becomes smaller and smaller, and noise grow to be dominant. In other words, $q(x_T|x_0) \simeq \mathcal{N}(x_T; 0, 1)$ holds for any $x_0$, so the final data $x_T$ is a complete noise. By using a normal distribution for the adding noise, the conditional probability at any time $t$ under a condition $x_0$ can be written in the following analytical expression:

$$q(x_t|x_0) = \mathcal{N}\left(x_t; \sqrt{\bar{\alpha}_t}x_0, \bar{\beta}_t\right), \tag{A.3}$$

with

$$\bar{\alpha}_t = \prod_{s=1}^{t} \alpha_s, \qquad \bar{\beta}_t = 1 - \bar{\alpha}_t. \tag{A.4}$$

The reproductive property of the normal distribution leads to this result.

Now, consider the complete noise $\mathcal{N}(x_T; 0, 1)$ as an initial value and follow the diffusion process in the reverse direction. This is called a reverse process. The conditional probability $p_\theta(x_{t-1}|x_t)$ at each step is estimated by a mathematical model characterized by parameters $\theta$. In this study, a neural network was used as the mathematical model. The whole Markov process is described as follows:

$$p_\theta(x_{0:T}) = p(x_T) \prod_{t=1}^{T} p_\theta(x_{t-1}|x_t), \tag{A.5}$$

$$p(x_T) = \mathcal{N}(x_T; 0, 1). \tag{A.6}$$

The probability distribution $p_\theta(x_{0:T})$ that generates a set of data $x_{0:T} = x_0, x_1, \ldots, x_T$ is called a likelihood. If $T$ is large and the increments of $\beta$ are sufficiently small, the diffusion process and the reverse process have the same functional form [35]. Therefore, $p_\theta(x_{t-1}|x_t)$ for each step follows the following equation:

$$p_\theta(x_{t-1}|x_t) = \mathcal{N}(x_{t-1}; \mu_\theta(x_t, t), \sigma_\theta(x_t, t)). \tag{A.7}$$

According to Ref. [36], a fixed variance $\sigma_\theta(x_t, t) = \sigma_t^2$, which is independent of the parameter $\theta$, ensures stability and accuracy of learning:

$$p_\theta(x_{t-1}|x_t) = \mathcal{N}(x_{t-1}; \mu_\theta(x_t, t), \sigma_t^2). \tag{A.8}$$

It is known that the same results can be obtained in the reverse process whether $\sigma_t^2 = \beta_t$ or $\sigma_t^2 = \bar{\beta}_t$ is chosen.

## A.2 Learning in diffusion process

When optimizing the parameters $\theta$ of the mathematical model with realistic computational costs, we often perform maximum likelihood estimation, which is formulated by maximizing the evidence lower bound of the log-likelihood. This is detailed below. The original input data $x_0$ is an observable variable, while the remaining $x_{1:T}$ is latent. Thus, in the reverse process, the likelihood $p_\theta(x_0)$ of the observed variable $x_0$ is obtained by integrating these latent variables as follows:

$$p_\theta(x_0) = \int dx_{1:T}\, p_\theta(x_{0:T}). \tag{A.9}$$

Since it is practically impossible to evaluate this integral, maximizing the likelihood itself is difficult. Therefore, we consider the following inequality that holds for the log-likelihood:

$$-\log p_\theta(x_0) \leq \mathbb{E}_{q(x_{1:T}|x_0)}\left[-\log \frac{p_\theta(x_{0:T})}{q(x_{1:T}|x_0)}\right] \tag{A.10}$$

$$= \mathbb{E}_{q(x_{1:T}|x_0)}\left[-\log \frac{p_\theta(x_0|x_1)p_\theta(x_1|x_2)\cdots p_\theta(x_{T-1}|x_T)p_\theta(x_T)}{q(x_T|x_{T-1})q(x_{T-1}|x_{T-2})\cdots q(x_1|x_0)}\right] \tag{A.11}$$

$$= \mathbb{E}_{q(x_{1:T}|x_0)}\left[-\log p_\theta(x_T) - \sum_{t\geq 1}\log \frac{p_\theta(x_{t-1}|x_t)}{q(x_t|x_{t-1})}\right] \equiv L(\theta). \tag{A.12}$$

In the transformation from Eq. (A.10) to Eq. (A.11), $p_\theta$ and $q$ are decomposed into products based on the Markov property of diffusion and reverse processes. In the end, maximum likelihood estimation for parameter updating is achieved by minimizing $L(\theta)$.

By rewriting $L(\theta)$ using Kullback-Leibler divergence[†]:

$$D_{\mathrm{KL}}(P\|Q) = \int dx\, P(x)\log \frac{P(x)}{Q(x)}, \tag{A.13}$$

---

[†]$D_{\mathrm{KL}}(P\|Q)$ is a quantified evaluation of the difference between the two probability distributions $P, Q$.

we obtain the following representation [36]:

$$L\left(\theta\right) = \mathbb{E}_q \Bigg[ \underbrace{D_{\mathrm{KL}}\left(q(x_T|x_0)\|p(x_T)\right)}_{L_T}$$

$$+ \sum_{t>1} \underbrace{D_{\mathrm{KL}}\left(q(x_{t-1}|x_t, x_0)\|p_\theta(x_{t-1}|x_t)\right)}_{L_{t-1}} - \underbrace{\log p_\theta(x_0|x_1)}_{L_0} \Bigg]. \tag{A.14}$$

$L_T$ is not related to the minimization of $L\left(\theta\right)$ because it does not include the parameter $\theta$. Regarding $L_0$, the last reverse process is considered a normal distribution that is independent for each dimension of the input data. Then, $L_0$ is evaluated by discretizing the interval $[-1, 1]$ into $k$ pieces as follows:

$$p_\theta(x_0|x_1) = \prod_{i=1}^{d} \int_{\sigma_-(x_0^i)}^{\sigma_+(x_0^i)} dx\,\mathcal{N}\left(x; \mu_\theta^i\left(x_1, 1\right), \sigma_1^2\right) \tag{A.15}$$

$$\sigma_+\left(x\right) = x + \frac{1}{k}, \quad \sigma_-\left(x\right) = x - \frac{1}{k}. \tag{A.16}$$

$L_{t-1}$ is evaluated as follows using a constant $C$ that is independent of $\theta$ and a noise $\epsilon$ following a standard normal distribution $\mathcal{N}\left(0, 1\right)$:

$$L_{t-1} - C = \mathbb{E}_{x_0,\epsilon} \left[ \frac{1}{2\sigma_t^2} \left\| \frac{1}{\sqrt{\alpha_t}} \left( x_t - \frac{\beta_t}{\sqrt{\bar{\beta}_t}}\epsilon \right) - \mu_\theta(x_t, t) \right\|^2 \right], \tag{A.17}$$

$$x_t = \sqrt{\bar{\alpha}_t}x_0 + \sqrt{\bar{\beta}_t}\epsilon. \tag{A.18}$$

To simplify the representation, a variable transformation is applied as

$$\mu_\theta(x_t, t) = \frac{1}{\sqrt{\alpha_t}} \left( x_t - \frac{\beta_t}{\sqrt{\bar{\beta}_t}}\epsilon_\theta \right). \tag{A.19}$$

Here, $\epsilon_\theta$ is predicted noise from the mathematical model. With this transformation, the right-hand side of Eq. (A.17) is summarized in the following form:

$$L_{t-1} - C = \mathbb{E}_{x_0,\epsilon} \left[ \frac{\beta_t^2}{2\sigma_t^2\alpha_t\bar{\beta}_t} \left\| \epsilon - \epsilon_\theta\left(x_t, t\right) \right\|^2 \right], \tag{A.20}$$

A rigorous evaluation of $L\left(\theta\right)$ is calculated by the sum of $L_T$, $L_{t-1}$, and $L_0$. In practical optimization, it is known that the following simplified function is sufficient [36]:

$$L_{\mathrm{simple}}\left(\theta\right) = \mathbb{E}_{t,x_0,\epsilon} \left[ \left\| \epsilon - \epsilon_\theta\left(x_t, t\right) \right\|^2 \right], \tag{A.21}$$

$$x_t = \sqrt{\bar{\alpha}_t}x_0 + \sqrt{\bar{\beta}_t}\epsilon. \tag{A.22}$$

In summary, the learning of DDPM proceeds in the following three steps. First, $x_t$ is constructed by adding noise to the initial data $x_0$ based on the noise schedules. Second, the noise $\epsilon$ added to the noisy data $x_t$ is estimated with the mathematical model. Finally, the parameters $\theta$ of the mathematical model are updated so that the difference between the actual noise $\epsilon$ and the predicted noise $\epsilon_\theta$ is reduced. These procedures are repeated until $L_{\mathrm{simple}}$ converges.

## A.3 Data generation in reverse process

During generating data through the reverse process, the perfect noise $x_T$ is used as initial data, and $x_{t-1}$ based on $x_t$ is sampled in sequence according to Eq. (A.8). This sampling method is called ancestral sampling. The average of the normal distribution is $\mu_\theta(x_t, t)$, which was estimated in Eq. (A.19). Thus, sampling from $p_\theta(x_{t-1}|x_t)$ corresponds to determining $x_{t-1}$ by the following equation:

$$x_{t-1} = \frac{1}{\sqrt{\alpha_t}} \left( x_t - \frac{\beta_t}{\sqrt{\bar{\beta}_t}} \epsilon_\theta \right) + \sigma_t u_t, \tag{A.23}$$

where $u_t$ is a disturbance following a standard normal distribution.

In summary, data generation by the reverse process of DDPM proceeds in the following two steps. First, the complete noise $x_T$ is used as initial data, and noise $\epsilon_\theta$ is predicted for each time using the mathematical model already trained in the diffusion process. Second, $\epsilon_\theta$ is removed from the data $x_t$ and some disturbance $u_t$ is added. The final result $x_0$ is obtained as new data by repeating these procedures.

## B   Conditional diffusion models

The success of the diffusion models is attributed not only to its ability to generate diverse data from complete noise but also to its capacity to produce preferred outputs based on arbitrary conditional labels. For instance, the diffusion models can generate images that correspond to any given text. Additionally, it can create high-resolution images from low-resolution inputs, and this process is known as super-resolution. To ensure smooth generation under various conditions, it is preferable to utilize a single trained model that can manage multiple labels, rather than retraining the model for each individual label.

In this section, we provide a brief introduction to classifier guidance (CG) [37], which enables conditional generation, and classifier-free guidance (CFG) [38], which further enhances the learning process. The following explanation is based on a formulation known as the score-based models (SBMs) [39, 40]. Although the derivations of SBM and DDPM are different, the final evaluation functions of both models are similar (Eq. (A.21)). It is established that optimal solutions for minimizing the evaluation functions are also consistent, and in most cases, both definitions can be applied as diffusion models [41]. Therefore, the derivation of SBM is not detailed in this paper.

## B.1 Classifier guidance

In a well-trained diffusion model, the gradient of the log-likelihood in the diffusion process is calculated as follows:

$$s(x_t, t) = \nabla_{x_t} \log q(x_t)$$
$$= -\frac{\epsilon_\theta(x_t, t)}{\sigma_t}. \tag{B.1}$$

This gradient $s$ is called a score. On the other hand, in general, conditional probability can be transformed based on Bayes' theorem:

$$q(x_t|c) = \frac{q(c|x_t)q(x_t)}{q(c)}. \tag{B.2}$$

Thus, given a label $c$ for a diffusion model, its conditional score is calculated as follows:

$$\begin{aligned} s(x_t, t, c) &= \nabla_{x_t} \log q(x_t|c) \\ &= \nabla_{x_t} \log q(c|x_t) + \nabla_{x_t} \log q(x_t) \\ &= -\frac{\epsilon_\theta(x_t, t)}{\sigma_t} + \nabla_{x_t} \log q(c|x_t) \\ &= -\frac{\epsilon_\theta(x_t, t) - \sigma_t \nabla_{x_t} \log q(c|x_t)}{\sigma_t}. \end{aligned} \tag{B.3}$$

In CG, $\nabla_{x_t} \log q(c|x_t)$ is estimated using a learned classification model $p_\theta(c|x_t)$. In other words, the new classification model (not the diffusion model) induces the generation of data that fits designated labels. Specifically, $\epsilon_\theta$ is corrected as follows:

$$\begin{aligned} \hat{\epsilon}_\theta(x_t, t, c) &= \epsilon_\theta(x_t, t) - \gamma \sigma_t \nabla_{x_t} \log q(c|x_t) \\ &= \epsilon_\theta(x_t, t) - \gamma \sigma_t \nabla_{x_t} \log p_\theta(c|x_t), \end{aligned} \tag{B.4}$$

and the reverse process depends on the corrected noise $\hat{\epsilon}_\theta$ instead of the simple noise $\epsilon_\theta$ to generate data that matches the label $c$. $\gamma > 0$ is called a guidance scale. $\gamma = 1$ matches the original conditional probability, and $\gamma > 1$ emphasizes the conditional label.

There are two practical issues with CG. The first is that it requires the score $\nabla_{x_t} \log p_\theta(c|x_t)$, which depends on time $t$. To estimate this score, a classification model $p_\theta(c|x_t)$ must be trained for each various noise level, so a computational cost is high. The second is that poor quality of conditional generation is realized when data $x$ and labels $c$ have little or no relation to each other.

## B.2 Classifier-free guidance

CFG does not use the classification model $p_\theta(c|x_t)$, but only the diffusion model to learn conditional scores directly. Specifically, a diffusion model that can receive the label $c$ is constructed, and the corrected noise $\hat{\epsilon}_\theta$ is transformed as follows:

$$
\begin{aligned}
\hat{\epsilon}_\theta\left(x_t, t, c\right) &= \epsilon_\theta\left(x_t, t\right) - \gamma \sigma_t \nabla_{x_t} \log q\left(c|x_t\right) \\
&= \epsilon_\theta\left(x_t, t\right) - \gamma \sigma_t \nabla_{x_t} \log \frac{q\left(x_t|c\right) q\left(c\right)}{q\left(x_t\right)} \\
&= \epsilon_\theta\left(x_t, t\right) - \gamma\left[\sigma_t \nabla_{x_t} \log q\left(x_t|c\right) - \sigma_t \nabla_{x_t} \log q\left(x_t\right)\right] \\
&= (1-\gamma)\,\epsilon_\theta\left(x_t, t\right) + \gamma \epsilon_\theta\left(x_t, t, c\right).
\end{aligned}
\tag{B.5}
$$

In other words, the corrected noise $\hat{\epsilon}_\theta$ is determined by the linear combination of the unlabeled noise and the labeled noise. The coefficient $\gamma$ is called the CFG scale, which was adopted as 8.0 in our analysis. The larger the CFG scale, the more faithful the data are to the label $c$, but the diversity of generated results tends to be lost. On the other hand, when $\gamma < 1$, it emphasizes the diversity.

At first glance, the evaluation of Eq. (B.5) require two mathematical models, $\epsilon_\theta(x_t, t, c)$ and $\epsilon_\theta(x_t, t)$. However, a common mathematical model can be used by using $\varnothing$ to denote the absence of conditional labels as $\epsilon_\theta(x_t, t) = \epsilon_\theta(x_t, t, c = \varnothing)$. For $\varnothing$, a learned embedding vector or a zero vector $\varnothing = 0$ is often used. In updating the parameter $\theta$, $c = \varnothing$ is adopted with low probability (10-20%), and the learning of CFG proceeds with mixing the cases with and without labels.

CFG does not require an independent classification model. It needs just dropping out the conditional part with a certain probability in usual learning. Therefore, the learning process with CFG can be significantly simplified in comparison to CG. Furthermore, the correspondence between data $x$ and labels $c$ can be predicted with good accuracy, so high-quality conditional generation can be realized.

## C  Transfer learning

In this section, we formulate the transfer learning. For reference, Ref. [42] provides an overview of transfer learning. The domain for preparing a learned neural network is referred to as the source domain, while the domain in which the same network is applied is called the target domain. When constructing a neural network $Y_s$ using input data $X_s$ from the source domain, a multilayer structure of the network is expressed as follows based on Eq. (2.11).

$$
Y_s = X_P \circ X_{P-1} \circ \cdots \circ X_1\left(X_s\right),
\tag{C.1}
$$

with a whole number of the layers $P$. Then, we focus on a submodel $\Phi = X_Q \circ X_{Q-1} \circ \cdots \circ X_1$ up to the $Q$ layers of the trained model $Y_s$, where $Q < P$. In a well-trained neural network in the source domain, parameters of $\Phi$ include features that are useful for predicting $Y_s$.

If there are common features between the source and target domains, $\Phi$ is expected to be useful for prediction in the target domain also. Thus, under input data $X_t$ from the target domain, training a model of the form $Y_t = f_t \circ \Phi(X_t)$ is called transfer learning through feature extraction. Here, $f_t$ represents an arbitrary neural network. Since $\Phi$ has already been trained on the source domain, its parameters remain fixed. In other words, since learning in the target domain is limited to $f_t$, there is no need to update all of the parameters of $Y_t$. Furthermore, transfer learning enables the construction of a neural network with high accuracy even with a limited amount of data $X_t$.

The fine-tuning used in this study is a type of transfer learning. In this approach, $Q = 0$ is adopted in the aforementioned formulation, and the parameters of the network $Y_s$ trained in the source domain are set as the initial values of $Y_t$. Then, $Y_t$ is retrained using the dataset $X_t$ from the target domain. While allowing all parameters to be updated ensures flexibility of learning in the target domain, a large amount of data $X_t$ should be prepared compared to the transfer learning with $Q > 0$.

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
