# Peer review of "Exploring the flavor structure of leptons via diffusion models"

_SciPost Physics_

## Round 1 · Referee Report · Anonymous (Referee 1) · 2025-5-17

Report

The authors introduce a diffusion generative model to study the flavour structure of leptons. In particular, the authors are interested in generating the neutrino mass matrix conditioned on experimental values. Through a fine-tuning process, the authors refine the training dataset, originally created with uniform priors for to-be generated objects, with a more efficient set that is more consistent with the experimental data. While the idea is interesting and novel, the discussion of the findings in this paper is based on events generated from the diffusion model without convincing evidence that the diffusion model is indeed working! Studies on the validity of the diffusion model and choice of prior should be performed if the authors want to claim any property of the generated events. See below for more detailed comments.

Are the uncertainties from the values measured in Table 1 correlated? If so, do you sample from the correct covariance matrix when generating the dataset G with the diffusion model from the labels? Similarly, when you say in the Results section that 104 solutions have all physical quantities within 3 sigma in L, do you assume any type of correlation between the experimental uncertainties?

I’m also confused with the efficiency of the diffusion model based on these numbers. I assume you generate the 3e7 G events by sampling from experimental values of L within the uncertainties, possibly drawn from a gaussian distribution? From those you say only 104 solutions are consistent with the values of L, which has a ~3e-5 efficiency, why is this the case? I would expect the generation, conditioned on L, to give a higher fraction of samples compatible with the L value used. Why is it not the case?

Perhaps it is my lack of understanding of the problem, but if we assume that each value of L is uniquely determined by a generated value G (even though multiple Gs can be associated to a single L, hence why this work is relevant), then the lower acceptance for generated values compatible with the condition L shows that the diffusion model is not working properly. Is my understanding incorrect?

What were the studies done to ensure the diffusion model is indeed learning what you think it is learning? For instance, is the first diffusion model generating uniform distributions in each feature of G when you first create the initial dataset for training? How does the gamma value for the classifier-free training change the performance/quality of generated samples?

Additionally, were any coverage tests performed? To me it is already worrisome that a small fraction of events are actually compatible with the initial values used, but for the validity of any subsequent study performed with this generated dataset, the authors should show that the diffusion model is indeed sampling correctly from the learned pdf distribution. You can simply show the pull for example, by showing that the chi2 distribution in 2.22 with the new values of Pl being normally distributed if the coverage is correct, is that the case?

Considering the hierarchy, the authors claim that the prior choice of the uniform distribution leads to a 2-digit hierarchy. Isn’t that a bad sign that the end model has a strong prior dependence and that generated values will strongly depend on the initial choice of prior?

My main worry in this case comes from the studies performed after, where the authors study the correlations for different observables calculated based on the values generated. If the initial prior dictates the generated values, then the correlations observed might simply be artifacts from the choice of prior. Worse, if the diffusion model is not accurate, then generated values might be biased towards non-existent correlations, rendering the following discussion incorrect.

Recommendation

Ask for major revision

---

## Round 1 · Referee Report · Anonymous (Referee 2) · 2025-6-4

Strengths

1-Use of state-of-the-art generative network

Weaknesses

1-Missing statistical foundation
2-Poor interpretability of the results
3-Missing citation to related work
4-Weak and inconsistent language
5-Poorly motivated

Report

The authors attempt to probabilistically invert experimental observations to recover theoretical parameters using generative machine learning. While similar inference approaches have been explored in other areas of high-energy physics, this is, to my knowledge, the first attempt to apply neural generative methods to the problem of neutrino flavor structure.
Specifically, the authors train a generative model to learn the conditional distribution of the type-I seesaw Lagrangian parameters G given the observed neutrino mixing matrix and mass differences L. They use a conditional diffusion model with classifier-free guidance, which is state-of-the-art in many domains, and adapt it to this physics task.

The training is done in two stages. First, the model is pretrained on data drawn from Gaussian and log-uniform priors over G covering a broad range of the parameter space. Then, the authors fine-tune the model on a subset of samples whose predicted observables L(G) lie close to the actual experimental value of L.

This two-stage setup is described as “transfer learning”, but in a strict machine learning sense, transfer learning involves training across different tasks or datasets. Here, the task and target distribution remain the same only the training data is modified. Hence, their phrasing could be misleading.

In addition, this fine-tuning step has an important statistical consequence as it alters the effective prior over G. After filtering, the model no longer samples from the original prior, and the resulting posterior is now conditioned on a non-analytical, empirically defined prior. Yet the authors continue to describe their outputs as posterior samples, without addressing how the posterior is defined or how the prior shift affects interpretability. These issues are not framed in any Bayesian statistical language, and no explicit probabilistic model is written. Terms like prior, posterior, or likelihood are used loosely or not at all. If this is meant as a form of neural Bayesian inference, which is I think the goal of the paper, then it should be made precise. Currently, this conceptual layer is missing.

In the results section, the authors state that they generated 30 Million G-samples from their trained diffusion network and found only 104 configurations that pass their closure test, i.e., for which the predicted L(G) lies within 3σ of the experimental L. This implies an acceptance rate of 0.00035%, which is very low. That suggests that the network is either poorly trained, collapsed, or misconditioned.

This exposes a deeper issue, the learned posterior is not validated statistically. While a closure test is performed, this merely confirms that some samples are consistent with observation, it says nothing about whether the full posterior is covered or correctly shaped. The authors should apply more rigorous tests such as trying to check for calibration and coverage on a test dataset, where the true posterior is known.

Besides, the authors fail to explain why this neural method is preferable to classical parameter estimation techniques already in use, such as MCMC-based scanning (which can be interfaced with a neural important sampler). These frameworks have been successfully used in many BSM and flavor inference problems and provide rigorous statistical outputs. Without any comparison, justification, motivation or benchmarking, it is unclear whether the proposed method offers meaningful advantages in speed, coverage, or interpretability and thus, whether the paper is relevant or not.

Lastly, the paper suffers from poor grammar, awkward phrasing, and missing punctuation, which hinder its readability. For example, sentences like “In our program, 0 is used as ∅” are not only irrelevant but actively confusing. A complete proofreading pass is necessary. Additionally, the authors omit almost all relevant citations in neural posterior estimation (e.g. arxiv:2003.06281, arxiv:1911.01429) and do not frame their methodology in the context of existing research.

In the current form, I cannot recommend to publish the paper. It would need a complete make-over, starting from the introduction that lacks motivation, the theory section where important statistical definitions are missing and the result section which does not cover all relevant aspects of their study.

Recommendation

Ask for major revision

  • validity: poor
  • significance: poor
  • originality: good
  • clarity: poor
  • formatting: good
  • grammar: below threshold

---

## Editorial Decision

awaiting_resubmission